# Study on the measurement of industrial eco-efficiency, spatial distribution and influencing factors in Yangtze River Economic Belt

**Liang Liu[1], Yuting Zhao[1], Yirui Yang[1], Shu Liu[1], Xiujuan Gong[1], Pan Jiang[1,2]***

**1** College of Economics and Management, Southwest University of Science and Technology, 621010, Mianyang, China, **2** College of Environment and Resources, Southwest University of Science and Technology, 621010, Mianyang, China

☯ These authors contributed equally to this work.
* jp510722@163.com

**Data Availability Statement:** All relevant data are within the paper and its Supporting Information files.

## Abstract

Nowadays, China insists on the strategy of ecological priority and green development, and the Yangtze River Economic Belt is a pioneering demonstration belt for the construction of ecological civilization in China. Promoting industrial ecological efficiency is of great significance to Chinese sustainable development and high-quality economic development. Based on the provincial panel data of 11 provinces and cities in the Yangtze River Economic Belt region from 2011 to 2020, we use the super-efficient slacks-based measure of efficiency SBM model to measure the industrial eco-efficiency of the Yangtze River Economic Belt, reveal the spatial differences in industrial eco-efficiency of different provincial units, and explore the industrial eco-efficiency. The results show that: (1) from the perspective of the Yangtze River Economic Belt as a whole, industrial eco-efficiency is on a stable upward trend, but the efficiency level is not high, and there is more room for improvement; (2) in terms of the spatial distribution of industrial eco-efficiency in the Yangtze River Economic Belt, there are significant differences, with industrial eco-efficiency in the downstream region being higher than that in the midstream region and the upstream region, and the lowest in the midstream region, while in terms of the spatial distribution pattern, the industrial eco-efficiency of 11 provinces and cities in the Yangtze River Economic Belt eco-efficiency has a significant positive spatial autocorrelation; (3) in terms of influencing factors, economic development level, factor endowment, and foreign investment can significantly influence industrial eco-efficiency in the Yangtze River Economic Belt. The study's result helps provide theoretical guidance and practical reference for promoting the green and ecological development of industries in the Yangtze River Economic Belt.

## 1 Introduction

The rise and fall of ecology are related to the human civilization, and ecological governance has become particularly important in the global context of the growing ecological crisis, the "after-effects" of industrial civilization. Since the 1970s, the economy of China has continued to multiply. Along with it, sloppy economic growth has caused numerous environmental

**Funding:** LL have been funded by the Sichuan Soft Science Research Program Projects (Project Numbers: 2023JDR0278 and 2022JDR0177) and the National Natural Science Foundation of China (Project Numbers: 72004188). In addition, LL was responsible for the design and review of the manuscript.

**Competing interests:** The authors have declared that no competing interests exist.

problems, such as excessive consumption of resources and ecological breakdowns. Since the 18th National Congress of the Communist Party of China first proposed building a beautiful China, the country's ecological civilization has achieved remarkable results. The report in the 19th National Congress of the Communist Party of China emphasized the need to accelerate the reform of the ecological civilization system. China is paying more and more attention to ecological development and sustainable development. At the same time, industry is the engine that drives economic development, and its degree of ecological development significantly impacts the ecological civilization construction of the whole society. Industrial eco-efficiency is an important indicator to measure the degree of regional industrial decolonization. Covering 11 provinces and cities including Shanghai, Jiangsu, Zhejiang, Anhui and Jiangxi, the Yangtze River Economic Belt occupies an essential role in the high-quality development of Chinese society, and the ecological development of its industries has a significant impact on the construction of ecological civilization in the whole Chinese society. In November 2020, General Secretary Xi Jinping stressed during his research in Nanjing that it is necessary to adhere to the implementation of the new development concept, promote the high-quality development of the Yangtze River Economic Belt, and make the Yangtze River Economic Belt become Chinese the main battlefield of ecological priority and green development.

In this context, most of the studies on eco-efficiency in the existing literature have been conducted at the micro-level and municipal scale, such as tourism [1,2], industry [3,4], agriculture [5], and city cluster [6,7]. Fewer studies and measurements have been conducted on industrialization and provincial eco-efficiency; in terms of eco-efficiency measurement methods such as the indicator method [8,9], 3D ecological footprint model [10], DEA model [5,11–14], SFA model [15], and EBM model [16], the research methods have become increasingly diversified. There is little literature on the calculation of eco-efficiency values from an industry perspective using the SBM model considering non-desired outputs. In general, there is a lack of research on industrial eco-efficiency in academia, and there is a lack of targeted exploration focusing on a particular region. Based on this, how to realize the ecological green integration of the Yangtze River Economic Belt, scientific measurement of the industrial eco-efficiency of the Yangtze River Economic Belt, analysis of its spatial and temporal differences, and investigation of its influencing factors are of great significance to promote the construction of ecological civilization and high-quality development of the Yangtze River Economic Belt.

Therefore, this paper aims at the macro context of global ecological governance, focuses on the term "industrial eco-efficiency of Yangtze River Economic Belt", and draws the research framework of this paper, as shown in Fig 1, with the logical analysis line of "data source— index system construction—regional characteristics—spatial distribution—influencing factors" as the logical analysis line. the DEA-SBM model is used to measure the industrial eco-efficiency of 11 provinces and cities in Chinese Yangtze River Economic Belt, the Moran's I index is used to analyze the spatial differences, and the Tobit regression model is used to explore the industrial eco-efficiency of the Yangtze River Economic Belt. On the one hand, it is expected to provide enlightening opinions for Chinese efforts in the field of global ecological governance; on the other hand, it responds to the strategy of ecological priority and green development and promotes the construction of ecological civilization in China, to provide lessons for the whole country, the rest of provinces and regions, and providing the theoretical basis for achieving industrial green and high-quality development in the Yangtze River Economic Belt.

## 2 Literature review

First of all, from the definition of the concept, industrial eco-efficiency [17] is evolved from the term "eco-efficiency", which is an important index to measure the degree of industrial ecology

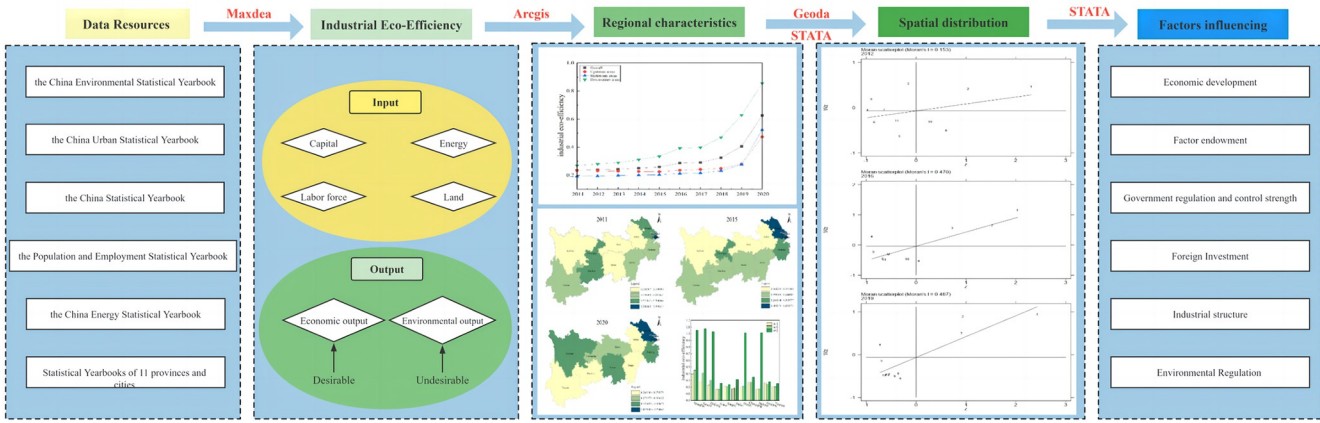

**Fig 1. Research framwork.**

and expresses the sustainable development of industry and the virtuous cycle of the ecosystem [18]. The research on industrial eco-efficiency at home and abroad starts from concept analysis and specific elaboration [19]. Scholars have defined the concept of industrial eco-efficiency in a rather heterogeneous way. Based on the concept of "eco-efficiency" put forward by the World Business Council for Sustainable Development (WBCSD) [20], many scholars have derived it and further expanded it from the macro [21] and micro [22] levels, summarizing industrial eco-efficiency as the degree of coordinated integration of economic, social and environmental ecosystems from both input and output perspectives. The core idea is to create the most economic and social output with the least amount of resource input and environmental cost [2].

Secondly, in terms of research content, scholars have conducted more studies on the eco-efficiency of specific industries, mainly including tourism [2,23], industry [3,4], agriculture [24], forestry [12], and energy [13], while fewer studies have been conducted on the eco-efficiency of the overall industry. Meanwhile, as the research continues to deepen, scholars have expanded their research from a single region to different regional scales, covering research on the whole country [25,26], geographical regions [27], provincial (city) [28,29], economic zones [30,31], and urban clusters [32,33]. Second, scholars have also explored in depth the specific effects and impacts of various factors on industrial eco-efficiency, such as industrial structure [29,34], fiscal decentralization [35], taxation [36], environmental regulation [37,38], technological innovation [14,39], and transportation structure [40].

In addition, from the perspective of research methods, the measurement methods of industrial eco-efficiency are also developing and improving, gradually changing from the single indicator method [8] and the indicator system method [9] to the model method. Among them, the single indicator method uses the ratio of economic output and environmental output to represent the eco-efficiency of different industries, and scholars generally choose income [39] to characterize economic output and carbon emission [37] to characterize environmental output. The indicator system method further takes into account the input elements of industries, but there are still some limitations in the variable assignment, so more and more scholars adopt the model method to measure industrial eco-efficiency. Among them, many domestic and foreign scholars choose the DEA model [41,42] to measure industrial eco-efficiency. Since the traditional DEA model does not consider non-desired outputs, thus scholars have made attempts such as the directional distance function method [43], but still have not solved the

slackness of inputs and outputs. Based on this, scholars gradually tend to use improved models such as SBM and super-efficient SBM to measure industrial eco-efficiency and analyze the spatial and temporal evolution trends and spatial spillover effects of industrial eco-efficiency by combining the nuclear density analysis method [34,44] and spatial Durbin model [45], while the driving factors of industrial eco-efficiency in specific sectors have been thoroughly explored by means of Tobit model [39,46] and SFA model [15,47]. Specifically, Haibo Chen (2020) assessed the regional eco-efficiency based on the super-efficiency DEA model with the gray entropy weight method [48]. Chang Xinfeng(2020) quantified eco-efficiency through the stochastic frontier model and pointed out that the industrial eco-efficiency of the Yangtze River Delta city cluster maintained a stable growth trend [49]. Zhang Xuxu(2022) used a non-expectation SBM model to measure the tourism eco-efficiency of the Yangtze River Delta city cluster from 2007–2017 and found that the overall efficiency of the tourism industry eco-efficiency of the city cluster was not high, and showed a trend of first decreasing and then increasing [50]. Wang Yaqin et al.(2021) used the super-efficient SBM model to measure the inter-provincial agricultural eco-efficiency from 2000–2018, and the results showed that: From 2000–2018, the overall agricultural eco-efficiency in China showed a stable upward trend, but still at a low level [51].

In summary, the research content and research methods on industrial eco-efficiency are becoming increasingly diversified. The research on industrial eco-efficiency in the Yangtze River Economic Belt is also gradually forming a complete system, but there are still the following shortcomings: first, there are more studies on industrial eco-efficiency in the Yangtze River Economic Belt at the municipal scale, while there is less literature on provincial differences; second, the method used ignores the influence of non-expected output on the results, which often results in certain measurement errors. Based on this, there are two main marginal contributions in this paper: first, the innovation of research perspective, which measures and analyzes the industrial eco-efficiency of 11 provinces and cities in the Yangtze River Economic Belt from the provincial perspective and selects the Yangtze River Economic Belt as the research object; second, the innovation of research method, which uses the super-efficient SBM model to measure and consider the non-expected output to reveal the distribution pattern of industrial eco-efficiency in different provincial units in the Yangtze River Economic Belt, and uses Moran's I index to analyze spatial differences, and finally explore the influencing factors of industrial eco-efficiency in the Yangtze River Economic Belt by establishing Tobit regression model, in order to provide a reference for decision making to comprehensively promote industrial eco-transformation in the Yangtze River Economic Belt.

## 3 Data and methodology

### 3.1 Sample and data sources

The Yangtze River Economic Belt is a globally influential inland economic belt spanning three eastern, central, and western China regions, covering 11 provinces and cities in Shanghai, Jiangsu, Zhejiang, Anhui, Jiangxi, Hubei, Hunan, Chongqing, Sichuan, Yunnan, and Guizhou. The area of the region is about 2,052,300 square kilometers, accounting for 21% of the Chinese land area and more than 40% of the country's population and gross domestic product [52]. In 2020, President Xi Jinping pointed out at the Symposium on Promoting the Development of the Yangtze River Economic Belt that it is necessary to adhere to the new development concept, promote the high-quality development of the Yangtze River Economic Belt, and make it become the main battlefield of ecological priority and green development in China. However, the Yangtze River Economic Belt, which is the "first demonstration belt for ecological civilization construction," is still facing many problems needed to be solved, mainly the deterioration

of the ecological environment, the difficult task of industrial upgrading, and the unbalanced regional development.

Considering the data availability, the data from 2011–2020 were selected for this study. Among them, industrial wastewater emissions, industrial $SO_2$ emissions, and industrial smoke and dust emissions are from the China Environmental Statistical Yearbook, GDP is from the China Statistical Yearbook, the employed population is from the Population and Employment Statistical Yearbook, built-up area data is from the China Urban Statistical Yearbook, energy consumption index data is from the China Energy Statistical Yearbook, and the rest of data is from Statistical Yearbooks of 11 provinces and cities in the Yangtze River Economic Belt from 2011 to 2020.

## 3.2 Methodology

**3.2.1 Super-efficient SBM model.** The traditional DEA model does not consider the non-consensual outputs, which can lead to deviations in the calculated efficiency values. Tone proposed the SBM model based on the traditional DEA model by considering input and output slack variables [53]. The super-efficient SBM model based on non-expected output has advantages in dealing with the non-expected output of industrial eco-efficiency and cross-sectional comparison of decision units. Compared with the traditional SBM model, the invalid DMU slack variables are further corrected to solve the problem of invalid efficiency ranking. The comparison of evaluation units with an efficiency value of 1 can also be solved. A more accurate evaluation of eco-efficiency under the condition of non-consensual output is achieved. Therefore, this paper uses the model to measure the industrial eco-efficiency of the Yangtze River Economic Belt by using 11 provinces and regions as DMUs.

The Super-SBM model considering non-desired outputs is as follows:

$$\gamma^* = min \frac{1 - \frac{1}{m}\sum_{i=1}^{m}\frac{s_i^x}{x_{i0}}}{1 + \frac{1}{q+r}(\sum_{p=1}^{q}\frac{s_p^y}{y_{p0}} + \sum_{k=1}^{r}\frac{s_k^b}{b_{k0}}}$$

$$\begin{cases} \sum_{j=1,j\neq 0}^{n} x_{ij}\lambda_j \leq x_{i0} + s_i^x (i = 1, 2, 3, \ldots, m) \\ \sum_{j=1,j\neq 0}^{n} y_{pj}\lambda_j \geq y_{p0} - s_p^y (p = 1, 2, 3, \ldots, q) \\ \sum_{j=1,j\neq 0}^{n} b_{kj}\lambda_j \leq b_{k0} + s_k^b (k = 1, 2, 3, \ldots, r) \\ \sum_{j=1}^{n}\lambda_j = 1 (VRS) \\ \lambda_J \geq 0, s_I^X \geq 0, s_p^y \geq 0, s_k^b \geq 0 \end{cases}$$

where, $s_i^x$, $s_p^y$ and $s_k^b$ are the slack variables of input-output, desirable output and non-desirable output, respectively, the amount of input and output that can be further optimized and improved; λ is the linear programming weight vector; VRS denotes variable payoff of scale; $x_{io}$ is the total input of the o-th decision unit to the i-th category of input, and the rest of the variables can be analogous; $\gamma^*$ is the industrial eco-efficiency value. Assuming that the production system has n homogeneous DMUs, each individual DMU utilizes m inputs and produces q desired outputs and r non-desired outputs, and the elements can be expressed as $x \in R^m$, $y \in R^q$,

$b \in R^r$. When $\gamma^* < 1$, the DMU belongs to the inefficient case, the input or output is not reasonable, when $\gamma^* \geq 1$, the DMU reaches the efficient.

**3.2.2 Spatial weighting matrix.** The spatial weight matrix is used to measure the degree of association between differing regions and to better fit the quality of industrial ecological development in the study area. The premise of data spatial measurement is to measure the spatial distance between regions. Define the spatial data from n regions as $\{x_i\}_{i=1}^n$, and the subscript i denotes region i. The distance between region i and region j is $w_{ij}$.

$$W = \begin{pmatrix} w_{11} & \cdots & w_{1n} \\ \vdots & \ddots & \vdots \\ w_{n1} & \cdots & w_{nn} \end{pmatrix}$$

The main diagonal element $w_{11} = \ldots = w_{nn} = 0$ in the above equation, where the spatial weight matrix W is a symmetric matrix. There are various methods to construct the matrix according to the adjacency criterion, and in this paper, the spatial weight matrix is constructed by using the Queen adjacency criterion through the analysis of Geoda software.

**3.2.3 Tobit model.** The Tobit model is an econometric method proposed by Tobin [54] for analyzing constrained explanatory variables. The industrial eco-efficiency of each province in the Yangtze River Economic Belt ranges from 0 to 1, and the data are restricted to a specific interval, which is a truncation problem. If we use OLS estimation, it will lead to biased estimation results. The Tobit model can avoid the errors caused by the general model, and the model expression is as follows:

$$Y = \begin{cases} Y^* = \beta X + \mu, \ Y^* > 0 \\ 0, \quad Y^* \leq 0 \end{cases}$$

Where Y represents the actual variable, $Y^*$ represents the restricted dependent variable, X represents the independent variable, β represents the regression parameter value, μ represents the error term, $\mu \sim (0, \sigma^2)$.

## 3.3 Construction of industrial eco-efficiency index system

The industrial eco-efficiency evaluation system is based on eco-efficiency and uses indicators to reflect the input and output according to the production function Y = F(K, L). Therefore, industrial eco-efficiency evaluation indexes should pursue the balance between industrial economic benefits and ecological and environmental impacts, considering not only the utilization efficiency of input factors in the industrial production process but also the ability to obtain desirable outputs, and at the same time, considering the environmental costs paid in production activities to obtain outputs. The research on the construction of industrial eco-efficiency index systems is generally evaluated from two aspects: environment and resources. The environmental aspect is mainly considered from the "three wastes"—waste gas, wastewater, and waste residue—discharged from industrial production. In contrast, the resource aspect is mainly considered from the direction of energy, capital, water, and land. In order to reflect the level of industrial eco-efficiency of the Yangtze River Economic Belt comprehensively, this paper combines the existing research based on the combing of the three primary input and output factors of economic factors, environmental impact factors, and natural resource factors, water, land, labor, energy, and capital are taken as input factors. Economic and environmental impact factors are taken as output indicators (as shown in Table 1).

**3.3.1 Input indicators.** Input indicators are used to measure the number of input factors. According to the production function, labor, capital, and land are selected as input indicators.

With the continuous development of industries, energy has become an important influencing factor for production activities in various industries. Thus energy is also used as an input indicator to make the measurement results of industrial eco-efficiency more effective. Among them, the labor force is expressed by the number of employed persons [55], which reflects the available labor force in a certain period; capital is measured by the amount of investment in fixed assets [56], which reflects the scale of investment in a region; the land is measured by the area of the built-up area [57], which indicates the land input of a region; energy is expressed by energy consumption [58], which reflects the energy input of a region.Water is measured by the total amount of regional water use, which mainly reflects the regional production, living and ecological water use in a certain period [4].

**3.3.2 Desirable output indicators.**   The desirable output indicator refers to the socio-economic benefits generated, meaning economic output. It is measured by gross domestic product (GDP), which reflects the final economic results of production activities in a certain period and is an important indicator to measure a region's economic status and development level. The lower the input, the higher the GDP reflects, and the higher the production efficiency of the region.

**3.3.3 Undesirable output indicators.**   Undesirable output mainly refers to the output of a bad environment caused by production activities. In this paper, industrial wastewater, industrial smoke, dust, and sulfur dioxide emissions are selected as non-desired output indicators to reflect the negative impact on the environment.

## 4 Analysis

### 4.1 Evaluation of industrial eco-efficiency

**4.1.1 Correlation analysis.**   In this paper, before calculating the industrial eco-efficiency of 11 provinces in the Yangtze River Economic Belt, a correlation analysis was conducted using STATA16. The correlation coefficient test results among the indicators are shown in Table 2. From Table 2, it can be seen that most of the indicators among GDP, industrial wastewater emissions, industrial emissions, industrial smoke and dust emissions, fixed asset investment amount, employment, energy consumption, and built-up area are most significant at 1%, and the indicators are strongly correlated. Among them, the correlation coefficient between energy consumption and industrial wastewater emission is 0.817, indicating that industrial wastewater emission and recycling strongly correlate with energy consumption.

**Table 1. Evaluation index of industrial ecological efficiency in Yangtze River Economic Belt.**

| Category | Index | Explaination | Unit of measure | Data sources |
|---|---|---|---|---|
| Input | Capital | Investment in fixed assets | Billion RMB | Statistical Yearbook of 11 Provinces and Cities in Yangtze River Economic Belt |
| | Labor force | Employed population | Ten thousand people | Statistical Yearbook of Population and Employment |
| | Energy | Energy consumption | Ten thousand tons of standard coal | China Energy Statistical Yearbook |
| | Land | Built up area | Square kilometer | China City Statistical Yearbook |
| | Water | Water consumption | billion cubic meters | China Environmental Statistical Yearbook |
| Desirable output | Economic output | GDP | Billion RMB | China Statistical Yearbook |
| Undesirable output | Environmental output | Industrial wastewater discharge | billion cubic meters | China Environment Statistical Yearbook |
| | | Industrial SO$_2$ emission | Ten thousand tons | China Environmental Statistical Yearbook |
| | | Industrial smoke and dust emission | Ten thousand tons | China Environmental Statistical Yearbook |

**Table 2. Correlation coefficient and *p* value of input—output indicators.**

| | Industrial eco-efficiency | GDP | Industrial wastewater emission | Industrial SO$_2$ emission | Industrial smoke and dust emission | Investment in fixed assets | Employed population | Energy consumption | Built-up area | Water consumption |
|---|---|---|---|---|---|---|---|---|---|---|
| Industrial eco-efficiency | — | | | | | | | | | |
| GDP | 0.579*** | 1.000 | | | | | | | | |
| Industrial wastewater emission | 0.166*** | 0.746*** | 1.000 | | | | | | | |
| Industrial SO$_2$ emission | -0.438*** | -0.224*** | 0.235** | 1.000 | | | | | | |
| Industrial smoke and dust emission | -0.336*** | 0.210** | 0.432*** | 0.445*** | 1.000 | | | | | |
| Investment in fixed assets | 0.321*** | 0.862*** | 0.576*** | -0.252*** | 0.358*** | 1.000 | | | | |
| Employed population | -0.017 | -0.615*** | 0.635*** | 0.227*** | 0.616*** | 0.728*** | 1.000 | | | |
| Energy consumption | 0.346*** | 0.898*** | 0.817*** | 0.114 | 0.375*** | 0.784*** | 0.785*** | 1.000 | | |
| Built-up area | 0.357*** | 0.934*** | 0.791*** | -0.053 | 0.361*** | 0.878*** | 0.932*** | 0.771*** | 1.000 | |
| Water consumption | 0.085 | 0.714*** | 0.734*** | 0.201 | 0.575*** | 0.735*** | 0.814*** | 0.796*** | 0.798*** | 1.000 |

"***", "**" and "*" indicate significant at the statistical levels of 10%, 5% and 1%, respectively.

That is, the treatment of energy consumption does not reduce the emission of industrial wastewater well in the Yangtze River Economic Belt area, the correlation coefficient between energy consumption and the amount of fixed asset investment is 0.784, which indicates that in the industrial ecological perspective, the investment in fixed assets is an essential reason for increasing energy consumption.

**4.1.2 Overall development characteristics. The trend of industrial eco-efficiency changes in the Yangtze River Economic Belt is shown in Fig 2.** The data of desirable output indicators, undesirable output indicators, and input indicators of 11 provinces and cities in the Yangtze River Economic Belt from 2011 to 2020 are substituted into the Super-SBM model through MaxDEA for calculation, and the results are shown in Table 3. Specifically, the development of industrial eco-efficiency in the Yangtze River Economic Belt can be analyzed from the following four perspectives.

From an overall perspective, the industrial eco-efficiency level of the 11 provinces included in the region is not high, with the highest value of eco-efficiency being 1.074 and the lowest value being 0.167, indicating that the overall industrial eco-efficiency level of the Yangtze River Economic Belt is not high and still has more rooms for improvement.

In terms of time, this paper takes 2011 as the starting point of the study and 2020 as the end point of the study due to the availability of data, with a period of 10 years. The industrial eco-efficiency of the Yangtze River Economic Belt shows an exponential trend of increasing yearly from 2011–2020. This trend indicates that before 2018, the productivity and economic development of the Yangtze River Economic Belt improved year by year, but the industrial wastewater, gas emissions, and energy consumption associated with the economic and industrial development did not improve, and insufficient attention was paid to the industrial ecology, which made the industrial eco-efficiency unable to get stable development in a certain period.

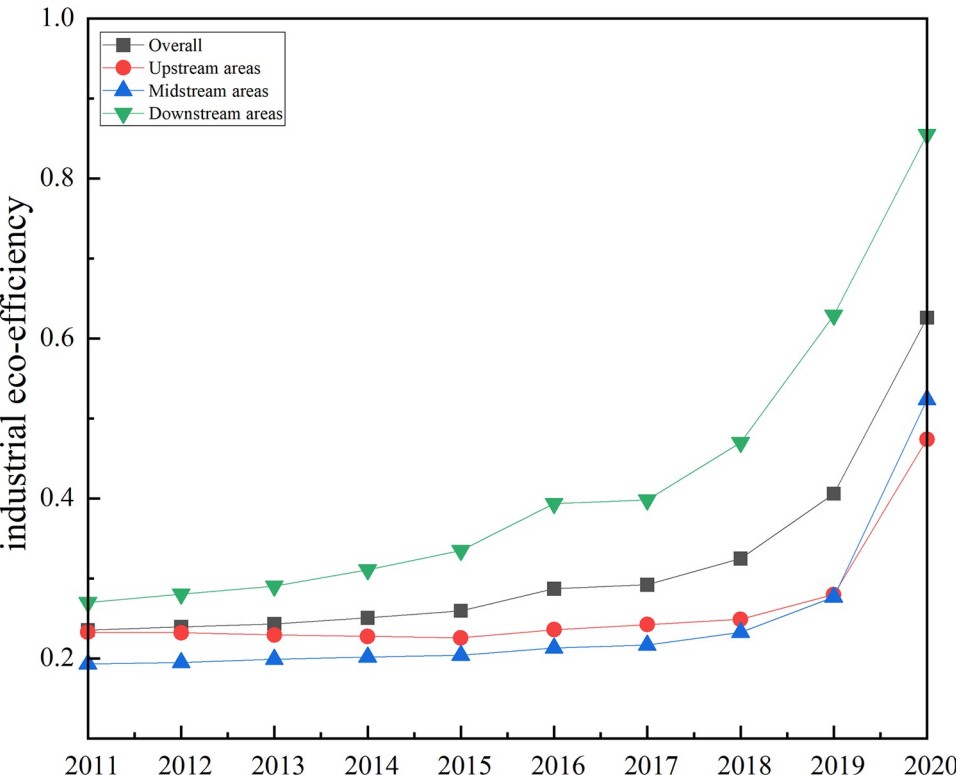

**Fig 2. Change trend of industrial eco-efficiency in the Yangtze River Economic Belt from 2011 to 2020.**

And then, after the 19th National Congress of the Communist Party of China proposed medicine to accelerate the reform of the ecological civilization system, the country increased the construction and investment in the ecological environment. Due to the natural resources consumed by industrial development and the impact on the environment with a certain time lag,

**Table 3. Evaluation results of industrial eco-efficiency of Yangtze River Economic Belt.**

| Province | 2011 | 2012 | 2013 | 2014 | 2015 | 2016 | 2017 | 2018 | 2019 | 2020 | Mean value |
|---|---|---|---|---|---|---|---|---|---|---|---|
| Shanghai | 0.399 | 0.410 | 0.397 | 0.420 | 0.453 | 0.542 | 0.414 | 0.572 | 1.017 | 1.051 | 0.568 |
| Jiangsu | 0.281 | 0.316 | 0.350 | 0.378 | 0.413 | 0.479 | 0.546 | 0.604 | 0.642 | 1.074 | 0.508 |
| Zhejiang | 0.231 | 0.228 | 0.247 | 0.274 | 0.305 | 0.378 | 0.448 | 0.512 | 0.634 | 1.036 | 0.429 |
| Anhui | 0.169 | 0.167 | 0.168 | 0.170 | 0.168 | 0.176 | 0.184 | 0.191 | 0.223 | 0.259 | 0.188 |
| Jiangxi | 0.213 | 0.215 | 0.212 | 0.213 | 0.210 | 0.216 | 0.221 | 0.222 | 0.232 | 0.240 | 0.219 |
| Hubei | 0.176 | 0.177 | 0.184 | 0.187 | 0.191 | 0.203 | 0.203 | 0.248 | 0.327 | 0.318 | 0.221 |
| Hunan | 0.190 | 0.192 | 0.200 | 0.205 | 0.211 | 0.220 | 0.227 | 0.228 | 0.271 | 1.013 | 0.296 |
| Chongqing | 0.276 | 0.284 | 0.281 | 0.281 | 0.275 | 0.294 | 0.309 | 0.300 | 0.316 | 0.350 | 0.297 |
| Sichuan | 0.171 | 0.173 | 0.176 | 0.175 | 0.177 | 0.180 | 0.195 | 0.231 | 0.299 | 1.014 | 0.279 |
| Guizhou | 0.268 | 0.260 | 0.248 | 0.243 | 0.244 | 0.264 | 0.252 | 0.249 | 0.260 | 0.278 | 0.257 |
| Yunnan | 0.215 | 0.211 | 0.211 | 0.211 | 0.206 | 0.206 | 0.214 | 0.216 | 0.245 | 0.253 | 0.210 |
| mean value | 0.235 | 0.239 | 0.243 | 0.251 | 0.260 | 0.287 | 0.292 | 0.325 | 0.406 | 0.626 | 0.316 |
| Upstream | 0.233 | 0.232 | 0.229 | 0.228 | 0.226 | 0.236 | 0.242 | 0.249 | 0.280 | 0.474 | - |
| Midstream | 0.193 | 0.195 | 0.199 | 0.202 | 0.204 | 0.213 | 0.217 | 0.233 | 0.277 | 0.524 | - |
| Downstream | 0.270 | 0.280 | 0.290 | 0.311 | 0.335 | 0.394 | 0.398 | 0.470 | 0.629 | 0.855 | - |

it was making the Yangtze River Economic Belt industrial eco-efficiency in the next three years. There was significant growth in industrial development at the same time, and the ecological environment has also been more effective. It also indicates that the industrial eco-efficiency of the Yangtze River Economic Belt is developing in a good direction.

From the three regions of the Yangtze River Economic Belt, the industrial eco-efficiency is in a stable trend, and the change trends of the upstream region, the midstream region, the downstream region, and the Yangtze River Economic Belt as a whole are synchronized, and all follow the overall change. Among them, the average value of industrial eco-efficiency in the downstream region of the Yangtze River Economic Belt, Shanghai, Jiangsu, Zhejiang, and Anhui provinces during 2011–2020 is higher than the average value of industrial eco-efficiency in the Yangtze River Economic Belt as a whole, while the upstream and midstream regions are lower than the Yangtze River Economic Belt as a whole, showing the trend of the upstream region in the downstream region in the midstream region. This phenomenon may be since the economic development level, environmental awareness, geographical location, and education level of the downstream region are better than those of the upstream and midstream regions, and the regional differences in industrial eco-efficiency still exist.

From the 11 provinces included in the Yangtze River Economic Belt, there are apparent differences in eco-efficiency levels among industries in each province. The top three industrial eco-efficiency values among the 11 provinces are Shanghai, Jiangsu, and Zhejiang, with Shanghai reaching 1.017 in 2019, Jiangsu and Zhejiang reaching 1.074 and 1.036 in 2020, respectively, while the bottom three industrial eco-efficiency rankings are Anhui, Jiangxi, and Yunnan. This may be because Shanghai, Jiangsu, and Zhejiang have developed economies, invested heavily in environmental protection, passed the high pollution stage of industrial development, and attracted much talent. In contrast, Anhui, Jiangxi, and Yunnan have slightly backward economies and do not pay enough attention to environmental management compared to other provinces, especially in the midstream region, Jiangxi Province, with high pollution. In particular, among the midstream regions, Jiangxi Province, as the critical zone for the transfer of high-pollution industries, has increased the pressure of local environmental management due to a large number of high-pollution enterprises moving in, thus causing the improvement of industrial eco-efficiency to stagnate.

**4.1.3 Regional characteristics.** The industrial eco-efficiency values of each province in the Yangtze River Economic Belt are divided into four levels to present better the spatial distribution pattern of industrial eco-efficiency of each province in the Yangtze River Economic Belt. This paper uses ArcGIS software to map the spatial distribution of industrial eco-efficiency of the Yangtze River Economic Belt in 2011, 2015, and 2020. The results are shown in Fig 3. From the spatial distribution pattern, there are significant geographical differences in the level of industrial eco-efficiency among the provinces in the Yangtze River Economic Belt. The industrial eco-efficiency of each province and city has changed to different degrees during the sample period. From an overall perspective, Shanghai, Jiangsu, and Zhejiang, located in the downstream region of the Yangtze River Economic Belt, always maintain their industrial eco-advantage and are in the first echelon, while Chongqing in the upstream region of the Yangtze River is the second. From the temporal level, in 2011, only Shanghai was located at a relatively high level of industrial eco-efficiency, followed by Chongqing and Guizhou, and followed by Zhejiang, Yunnan, and Jiangxi; in 2015, Jiangsu rose to the first echelon, Zhejiang rose to the second echelon, and Guizhou fell to the third echelon, and the trend of concentrating provinces with higher levels of industrial eco-efficiency in the downstream area of the Yangtze River Economic Belt became more and more pronounced, and the overall Yangtze River Economic Belt The industrial eco-efficiency level has improved; in 2020, Hunan Province rises to the first echelon, and the industrial eco-efficiency level becomes higher, Shanghai,

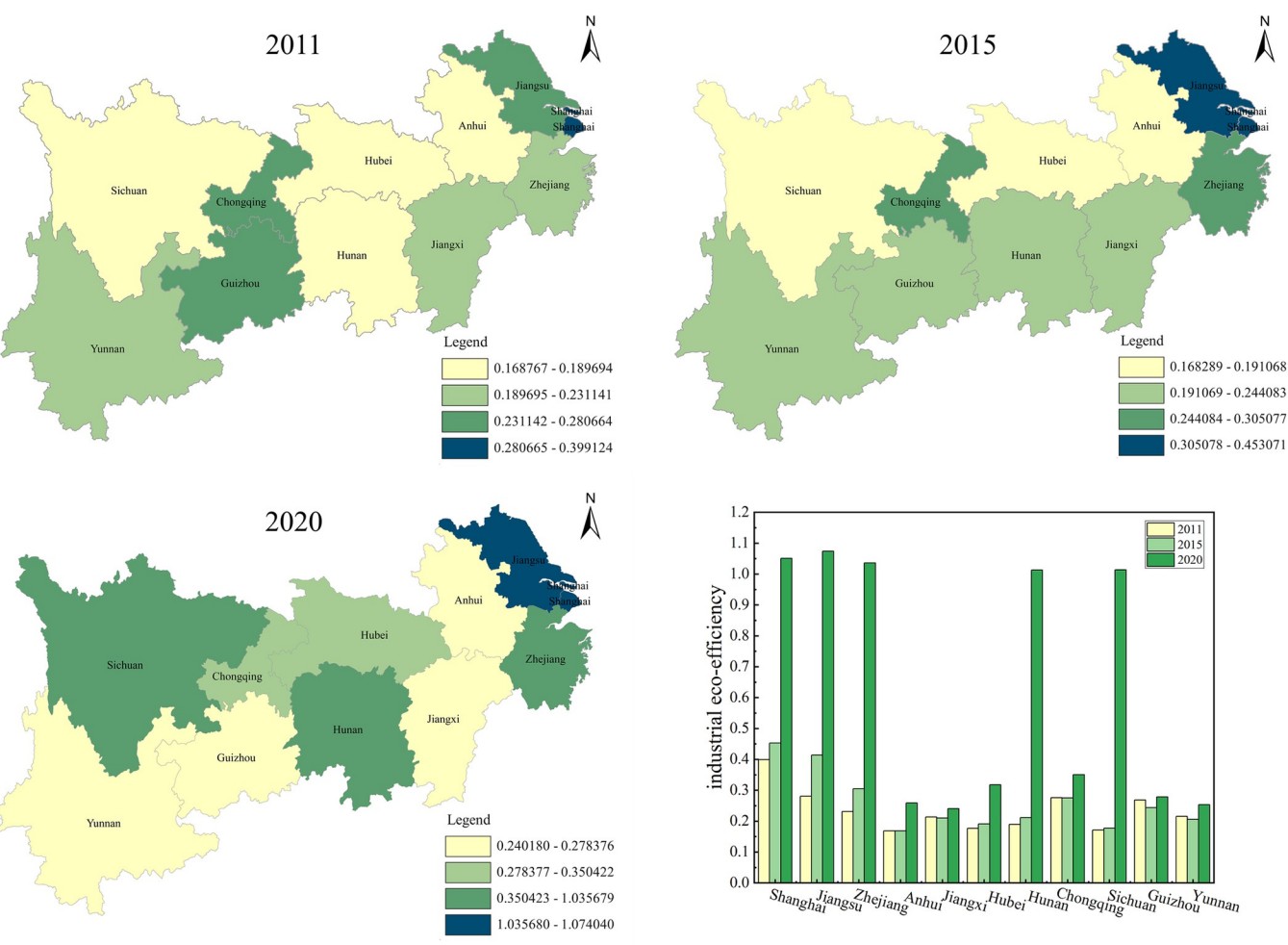

**Fig 3. Regional characteristics of industrial eco-efficiency.**

Zhejiang, and Jiangsu have efficiency values concentrated between 1.036–1.074, while Anhui Province's industrial eco-efficiency is still located at a low level. As shown by Fig 2 the spatial trend of industrial eco-efficiency in the Yangtze River economyis becoming more and more significant.

## 4.2 Spatial distribution

According to the above analysis, it can be found that there are apparent differences in industrial eco-efficiency among 11 provinces and cities in the Yangtze River Economic Belt, which indicates that industrial eco-capacity has a strong connection with spatial distribution. In particular, from a regional perspective, the industrial eco-efficiency in the upstream, midstream, and downstream regions shows a spatial distribution pattern with the highest in the downstream and the lowest in the midstream. Therefore, this paper uses global spatial autocorrelation and local spatial autocorrelation based on the study of industrial eco-efficiency in the Yangtze River Economic Belt to analyze the spatial distribution characteristics of industrial eco-efficiency among the 11 provinces.

**4.2.1 Global spatial autocorrelation.** Global spatial autocorrelation is used to analyze the spatial distribution characteristics of the 11 provinces and cities in the Yangtze River

Economic Belt within the sample period, which can reflect the correlation and dependency between each province and city. In this paper, we used STATA 16 to calculate the global Moran's I index of industrial eco-efficiency values of each province and city in the Yangtze River Economic Belt for each year and selected the spatial adjacency weight matrix with the existence of a common boundary [59], and conducted significance tests on all Moran's I indices, as shown in Table 4. It can be seen that the global Moran's I indexes of industrial eco-efficiency of the 11 provinces in the Yangtze River Economic Belt from 2011 to 2020 are all positive. However, the years 2011 and 2020 do not pass the significance test at the 10% statistical level, which means that the industrial eco-efficiency of the Yangtze River Economic Belt in these two years is scattered and distributed, with the middle of each year from 2012 to 2019 passed the significance test at the 10% statistical level for the agglomeration pattern. There was a significant positive spatial correlation. The global Moran's I only represent the trend of overall spatial correlation and is inadequate in reflecting the characteristics of local spatial distribution. Based on this, this paper uses local spatial autocorrelation to explore further the local correlation of the Yangtze River Economic Belt.

**4.2.2 Local spatial autocorrelation.** Local spatial autocorrelation can reflect the type of agglomeration of each province in the Yangtze River Economic Belt and the spatial correlation between this province and the neighboring adjacent provinces. To explore the spatial agglomeration characteristics within the economic belt, 2012, 2016, and 2019 were selected to draw local Moran's I scatter plots.

Each quadrant of Moran's I scatter plot represents a spatially correlated pattern. Specifically, provinces in the first quadrant belong to the "high-high" type (H-H), reflecting the continued growth or smooth change of industrial eco-efficiency between the province and the surrounding provinces. Provinces in the second quadrant belong to the "low-high" type (L-H), reflecting a lower level of industrial eco-efficiency than neighboring provinces. The provinces in the third quadrant are of the "low-low" type (L-L), reflecting the low level of industrial eco-efficiency of the province and its neighboring provinces. The provinces in the fourth quadrant are of the "high-low" type (H-L), reflecting the low level of industrial eco-efficiency of the province and its neighboring provinces. Provinces in quadrants one and three have similar industrial eco-efficiency levels and are characterized by solid spatial aggregation and positive correlation. In contrast, provinces in quadrants two and four have different industrial eco-efficiency levels from their neighboring provinces. They are characterized by substantial spatial divergence and negative correlation, meaning that when the province's industrial eco-efficiency level is weak or strong, the industrial eco-efficiency level of the neighboring provinces is stronger or weaker.

**Table 4. Statistical value of Moran's I of industrial eco-efficiency in the Yangtze River Economic Belt.**

| Year | Moran's I | Z-value | p-value |
|------|-----------|---------|---------|
| 2011 | 0.104 | 1.205 | 0.114 |
| 2012 | 0.153 | 1.438 | 0.075 |
| 2013 | 0.254 | 1.910 | 0.028 |
| 2014 | 0.340 | 2.365 | 0.009 |
| 2015 | 0.413 | 2.755 | 0.003 |
| 2016 | 0.470 | 3.030 | 0.001 |
| 2017 | 0.400 | 2.640 | 0.004 |
| 2018 | 0.533 | 3.238 | 0.001 |
| 2019 | 0.487 | 3.512 | 0.000 |
| 2020 | 0.003 | 0.488 | 0.313 |

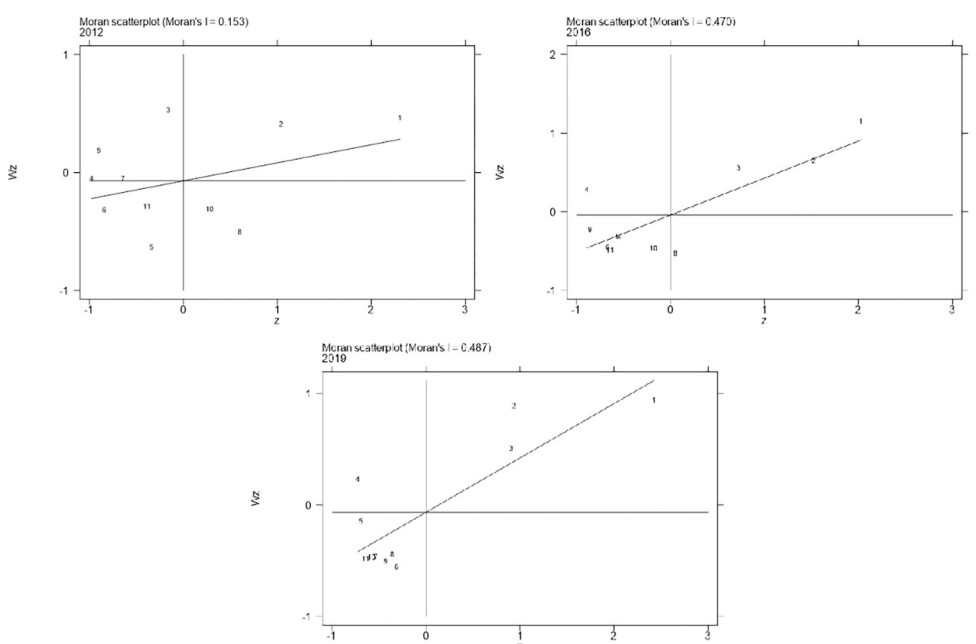

**Fig 4. Partial Moran's I scatter diagram.**

Fig 4 shows the distribution of the 11 provinces in the Yangtze River Economic Belt in each quadrant in 2012, 2016, and 2019. It can be observed that five provinces showed a positive spatial correlation in 2012, accounting for 45.45% of the overall, including two provinces in the first quadrant (H-H) and three provinces in the third quadrant (L-L). In 2016, nine provinces showed positive spatial correlation, accounting for 81.82% of the totality, including three provinces in the first quadrant (H-H) and the third quadrant (L-H) containing six provinces. Ten provinces showed a positive spatial correlation in 2019, accounting for 90.91% of the totality, including three provinces in the first quadrant (H-H) and seven provinces in the third quadrant (L-L). As a whole, which shows the leap state of the provinces in the quadrant of the Yangtze River Economic Belt, and the number of provinces showing positive spatial correlation has been increasing during the sample period, taking 2016 as the base point, the industrial eco-efficiency of the Yangtze River Economic Belt increased by 36.37% in the rate of positive spatial correlation from 2012 to 2016, and from 2016 to 2019 increased by another 9.09%. Indicating that the provinces in the Yangtze River Economic Belt follow the macro guidance of the Chinese 13[th] Five-Year Plan, implement the concept of green development, and pay attention to as well as carry out relevant work according to each province's situation, resulting in a gradual increase in the level and correlation of industrial eco-efficiency. The weight increase can reflect the poor local spatial stability of industrial eco-efficiency. However, with the elapse of time, provinces located in quadrants two and three gradually leap to quadrants one and three, which further indicates that the trend of heterogeneity of industrial eco-efficiency in the Yangtze River Economic Belt has decreased, while the level of homogeneity of efficiency has been increasing.

Remarks: 1: Shanghai, 2: Jiangsu, 3: Zhejiang, 4: Anhui, 5: Jiangxi, 6: Hubei, 7: Hunan, 8: Chongqing, 9: Sichuan, 10: Guizhou, 11: Yunnan

High-High Agglomeration Area (H-H). Shanghai, Jiangsu, and Zhejiang mainly represent the downstream area of the Yangtze River Economic Belt. It indicates that the industrial eco-efficiency of this region and the neighboring provinces show high and growing clustering

characteristics. The reason for this phenomenon may be that Shanghai, Jiangsu, and Zhejiang have a high level of economic development, well-developed transportation hubs, and effective measures and green ecological consciousness taken in paying attention to the Yangtze River ban, ecological, environmental protection and restoration, and focusing on green development, which not only promote the high level of development of their industrial eco-efficiency but also have a better diffusion effect on the surrounding neighboring areas.

Low-High Agglomeration Area (L-H). The representative province is Anhui, which indicates that the industrial eco-efficiency in Anhui province is low. However, the neighboring regions such as Shanghai, Jiangsu, and Zhejiang have higher levels of industrial eco-efficiency. The reason for this phenomenon may be that Anhui Province has a significant gap in economic development compared with the other three provinces in the lower reaches of the Yangtze River, a weak economic foundation, a relatively backward ecological civilization, and the traditional production methods of high pollution, high consumption and high emissions in this region cause severe ecological problems, which leads to a continuous low level of industrial eco-efficiency in Anhui Province and a low-high agglomeration area all the time during the sample period.

Low-Low Agglomeration Area (L-L). Mainly dominated by the upstream and midstream areas of the Yangtze River Economic Belt, such as Jiangxi, Yunnan, and Hunan, these areas have the disadvantage of having their low industrial eco-efficiency and the low level of industrial eco-efficiency in the neighboring regions. The reasons for this phenomenon may include the irrational development of the region, emphasis on logging, and disregard for environmental protection, especially in the upstream areas where soil erosion is severe and species in the Yangtze River are on the verge of extinction. The midstream areas are affected by subtropical monsoon climate subject to frequent flooding, hardening of shore slopes, deterioration of water quality, and severe water pollution, making the ecological environment less controllable. In terms of ecological resource utilization, a strong linkage is not achieved, making industrial eco-efficiency somewhat fragile.

High-Low Agglomeration Area (H-L). Chongqing and Guizhou mainly represent the upper reaches of the Yangtze River Economic Belt. Its industrial eco-efficiency is high, but the industrial eco-efficiency of neighboring provinces is low. Possible reasons are that Chongqing plays a pivotal role in Chinese economic development and western development. Guizhou is located in the southwest, has a large population base, is relatively backward in economic development, and has significant differences in resource endowments with other neighboring provinces, making it challenging to present a high level of agglomeration in a short period.

**4.2.3 Tobit regression.** Several factors jointly influence the industrial eco-efficiency of China's Yangtze River Economic Belt. The Super-SBM model can only measure the industrial eco-efficiency level and change trends from a static perspective. The efficiency values measured by this method are all greater than 0, which has a truncated tail [60], and a consistent estimate cannot be obtained using traditional least squares regression [61]. Therefore, this paper uses Tobit regression following the maximum likelihood method to summarize the critical influencing factors in previous studies [7,62–66].

1. Economic development, expressed by the ratio of local GDP to population. The higher the level of economic development of a city, the better the macroeconomic environment, which is more conducive to the ecological development of the industry;

2. Factor endowment, expressed as the ratio of local fixed asset investment to population;

3. Government regulation and control, expressed as the proportion of local fiscal expenditure to GDP. It is used to determine the impact of government support on industrial eco-efficiency;

**Table 5. Index of influencing factors of industrial eco-efficiency in the Yangtze River Economic Belt.**

| Variable | Code | Definition | Characteristic | Unit |
|---|---|---|---|---|
| Economic Development | ED | Local GDP as a share of population | + | % |
| Factor endowment | FE | Local fixed asset investment as a share of population | - | % |
| Government regulation and control strength | GRC | Local fiscal expenditure as a share of GDP | + | % |
| Foreign Investment | FI | Local foreign direct investment as a share of GDP | - | % |
| Industrial structure | IS | Local industrial value-added as a share of GDP | - | % |
| Environmental Regulation | ER | The proportion of industrial pollution investment treatment to the value added of secondary industry | + | % |

4. Foreign investment, expressed as the ratio of local FDI to GDP. The impact of foreign investment on industrial eco-efficiency The advantages and disadvantages of foreign investment on regional green development are still controversial, including the "pollution halo" hypothesis, which suggests that foreign investment can compensate for capital deficiencies and promote industrial upgrading, thus contributing to industrial eco-development, while some scholars argue that developed countries prefer to transfer highly polluting industries to Some scholars believe that developed countries prefer to transfer highly polluting industries to developing countries, thus increasing pollution and not taking advantage of the improved eco-efficiency of industries.

5. Industrial structure, expressed as the ratio of local industrial value added to GDP. The ratio of the industry-based secondary industry requires a large amount of energy input and emits a large amount of polluting waste such as wastewater, exhaust gas, and sulfur dioxide, thus the share of industry in the industry can have an important impact on the degree of decolonization of the regional industry.

6. Environmental regulation, expressed as the proportion of industrial pollution control investment in the value added of the secondary industry. According to Porter's hypothesis, reasonable environmental regulation is conducive to promoting enterprises' innovative technology, thus saving energy consumption, reducing industrial pollutant emissions, and facilitating the development of industrial ecology. For details, see Table 5 Industrial eco-efficiency impact factor indicators.

All data were obtained from the China Statistical Yearbook, China Environmental Statistical Yearbook, China Labor Statistical Yearbook, and statistical yearbooks of 11 provinces and cities. To compare the effects among the factors, six Tobit models were constructed by adding variables in the order of economic development (ED), factor endowment (FE), government regulation strength (GRC), foreign investment (FI), industrial structure (IS) and environmental regulation (ER), respectively.

$$IEE_{jt} = \alpha_0 + \alpha_1 ED_{jt} + \varepsilon_{jt} \qquad (I)$$

$$IEE_{jt} = \alpha_0 + \alpha_1 ED_{jt} + \alpha_2 FE_{jt} + \varepsilon_{jt} \qquad (II)$$

$$IEE_{jt} = \alpha_0 + \alpha_1 ED_{jt} + \alpha_2 FE_{jt} + \alpha_3 GRC_{jt} + \varepsilon_{jt} \qquad (III)$$

$$IEE_{jt} = \alpha_0 + \alpha_1 ED_{jt} + \alpha_2 FE_{jt} + \alpha_3 GRC_{jt} + \alpha_4 FI_{jt} + \varepsilon_{jt} \qquad (IV)$$

$$IEE_{jt} = \alpha_0 + \alpha_1 ED_{jt} + \alpha_2 FE_{jt} + \alpha_3 GRC_{jt} + \alpha_4 FI_{jt} + \alpha_5 IS_{jt} + \varepsilon_{jt} \tag{V}$$

$$IEE_{jt} = \alpha_0 + \alpha_1 ED_{jt} + \alpha_2 FE_{jt} + \alpha_3 GRC_{jt} + \alpha_4 FI_{jt} + \alpha_5 IS_{jt} + \alpha_6 ER_{jt} + \varepsilon_{jt} \tag{VI}$$

Where, j represents province; t represents year; $IEE_{jt}$ represents regional industrial eco-efficiency; $\alpha_0$ represents constant term; $\alpha_i$ represents regression coefficient (i = 1,2,. . .,6); $\varepsilon_{jt}$ is random disturbance term.

Before doing the sample regression analysis, the original data of each variable was tested for multicollinearity. The result was that the maximum VIF value was 4.75, and the average VIF was 2.83, less than 10. Therefore, there was no multicollinearity, and the variables were logged to reduce the influence of heteroskedasticity. The regression results are shown in Table 6.

From the regression analysis results in Table 6, it can be concluded that: firstly, the coefficients of economic development level in all six models are significant at the statistical level of 1%, and the regression coefficients are positive, which indicates that the economic development level can positively promote the industrial eco-efficiency of the Yangtze River Economic Belt, and the degree of influence is the greatest, which is consistent with the actual situation of the industrial eco-efficiency of the Yangtze River Economic Belt. The whole economic zone has a unique geographical advantage of excellent development potential. The gross product accounts for 40% of the country by improving the economic level, thus increasing the investment in environmental protection and solving the regional ecological environment. Secondly, the coefficient of factor endowment in Model II-Model VI is significant at the statistical level of 1% and plays a negative role, which means that the Yangtze River Economic Belt relies only on fixed asset investment and cannot effectively improve industrial eco-efficiency and may even have a suppressive effect. Although the construction and acquisition of fixed assets such as buildings and machines can significantly improve the development of the region, the construction of buildings and the operation of machines may be accompanied by the production of a large number of pollutants, which affects the regional ecological environment and reduces the industrial eco-efficiency. Third, the coefficients of foreign investment in Model IV-Model VI are significant at 1%, and 10% of the statistical level are significant and negative indicators. However, the coefficients are always less than 1, indicating that the degree of influence of foreign investment on the industrial eco-efficiency of the Yangtze River Economic Belt is not significant, which is consistent with the actual situation of the industrial eco-efficiency of the Yangtze River Economic Belt.

**Table 6. Regression analysis results of Tobit model.**

| Variable | Model I | Model II | Model III | Model IV | Model V | Model VI |
|---|---|---|---|---|---|---|
| Economic Development | 7.465*** | 8.361*** | 9.308*** | 9.780*** | 8.978*** | 8.971*** |
| Factor endowment | | -1.784*** | -1.701* | -2.537*** | -2.662*** | -2.651*** |
| Government regulation and control strength | | | 0.243** | 0.094 | -0.046 | -0.047 |
| Foreign Investment | | | | -0.114*** | -0.085* | -0.085* |
| Industrial structure | | | | | -0.345 | -0.347 |
| Environmental Regulation | | | | | | 0.002 |
| Constant term | -19.068*** | -17.006*** | *** | -18.975*** | -17.181*** | -17.184*** |
| Log-likelihood value | -26.378 | -24.521 | -22.333 | -18.817 | -18.243 | -18.242 |

"***", "**" and "*" indicate significant at the statistical levels of 10%, 5% and 1%, respectively.

## 5 Discussion

Regarding the overall development characteristics, the industrial eco-efficiency level of the 11 Yangtze River Economic Belt provinces is low during the sample period, with the average efficiency value around 0.32. Although the industrial eco-efficiency level has shown an upward trend in recent years compared with the pre-sample period, there is still much room for improvement overall. It is similar to the findings of Zhang (2022) [50], who argues that the steady increase in industrial eco-efficiency in the Yangtze River Economic Belt from 2010 to 2018 is due to the shift from high-speed socio-economic development to high-quality development, and in the stage of ecological civilization construction and green development, regional governments began to pay attention to ecological environmental protection and improve resource utilization efficiency thus promoting the improvement of industrial eco-efficiency in each region. The difference in industrial eco-efficiency levels among provinces is relatively apparent, the average value of industrial eco-efficiency in downstream areas is higher than that in midstream and upstream areas, which is the same conclusion as the research of Zhang (2022) [67]. Provinces with high industrial eco-efficiency levels, such as Shanghai, Jiangsu, and Zhejiang, are concentrated in economically developed downstream areas. In contrast, provinces with low industrial eco-efficiency levels, such as Yunnan, Jiangxi, and Hubei, are primarily concentrated in upstream and midstream areas. It may be due to the following factors: first, the economic development level, environmental awareness, geographical location, and education level of the downstream region are better than those of the upstream and midstream regions, thus leading to regional differences in industrial eco-efficiency in the Yangtze River Economic Belt; second, the midstream region of the Yangtze River is in the rising stage of industrial development and is the preferred place for the transfer of highly polluting industries from eastern China, and the transfer of industrial enterprises into the region has increased the burden on the local ecological environment, thus leading to lower industrial eco-efficiency [50].

The spatial distribution pattern shows a significant positive spatial autocorrelation of industrial eco-efficiency among the 11 provinces in the Yangtze River Economic Belt. Specifically, Moran's I value fluctuates steadily at the low value in the early period and at the high value in the later period, indicating that the industrial eco-efficiency of the Yangtze River Economic Belt shows a rising and falling trend during the sample period. The development among provinces is characterized by instability and variation. The research of Chang (2020) [49] also supports this argument by showing that the spatial distribution pattern of eco-efficiency development in the Yangtze River Delta urban agglomeration has significant regional differences and a "Z" shape distribution. The local Moran's I scatter plot shows the leap in the quadrant for each province in the Yangtze River Economic Belt, with an increasing number of spatially positively correlated provinces over the sample period. In general, the positive spatial correlation keeps strengthening. As the Chinese green concept of sustainable development is gaining popularity, the new urbanization continues to advance and the reform of the ecological civilization system deepens, the consequent emphasis on eco-environmental protection brings positive promotion effects to industrial development, provinces in quadrants two and three gradually move to quadrants one and three, and the spatial homogeneity of industrial eco-efficiency increases, showing a particular aspect of "catch-up effect" [68], which also means that provinces with low levels of efficiency get more efficient levels of regional influence to adopt a balanced development path focusing on the regional economy and ecological environment [49].

In terms of the influencing factors, the level of economic development has the greatest impact on the industrial eco-efficiency of the Yangtze River Economic Belt [67], it positively

promotes the industrial eco-efficiency, government regulation and control strength significantly enhances the industrial eco-efficiency of the Yangtze River Economic Belt, while factor endowment and foreign investment negatively inhibit the regional industrial eco-efficiency, which is in line with the actual development status of the Yangtze River Economic Belt, indicating that as the regional economic development level increases, the government and the public are more aware of the ecological environment This view is the same as that of Li (2021) [69], and in his research also concludes that industrial structure and foreign investment play a negative role in industrial eco-efficiency, which argues is because on the one hand, the increase in industrial structure means the increase in the proportion of regional polluting industries, which obviously reduces industrial eco-efficiency; on the other hand, China as a developing country, the more foreign investment, the more it causes scale and and structural effect [70], which results in pollution refuge effect; while Zhang (2022) [50] argues that the significant contribution of economic development level indicates that the Yangtze River Economic Belt pursues regional green development while maintaining economic growth, thus improving the energy saving and environmental protection technology and management in the production process, which provides the prerequisites for industrial eco-efficiency improvement.

# 6 Conclusions and recommendations

## 6.1 Conclusions

Using the Super-SBM model and Tobit model, this paper measures and analyzes the industrial eco-efficiency of each provincial administrative region in the Yangtze River Economic Belt based on the panel data of each province and city from 2011–2022, using 11 provinces and cities within the Yangtze River Economic Belt as the research object. Meanwhile, with the help of ArcGIS and STATA 16 software, the temporal evolution trend and spatial distribution pattern of industrial eco-efficiency in the Yangtze River Economic Belt were analyzed, and the conclusions of the study are as follows:

First, from the overall perspective, the industrial eco-efficiency of the Yangtze River Economic Belt shows a stable upward trend, but the efficiency level is not high, and there are significant differences in the efficiency level among provinces, which is specifically shown as downstream region > midstream region > upstream region.

Second, in terms of spatial distribution, there is a significant positive spatial autocorrelation in the industrial eco-efficiency of each province in the Yangtze River Economic Belt, which shows a leap in the quadrant, implying that the number of provinces demonstrating positive spatial correlation in the industrial eco-efficiency of the Yangtze River Economic Belt is increasing within the sample period again.

Third, from the perspective of influencing factors, among the six variables selected in this paper, the level of economic development and government regulation significantly improve the industrial eco-efficiency of the Yangtze River Economic Belt, among which the level of economic development has the most significant influence. In contrast, factor endowment and foreign investment do not effectively improve industrial eco-efficiency and may even have inhibitory effects.

## 6.2 Recommendations

In order to promote the development of green and ecological industries in the Yangtze River Economic Belt and, at the same time, effectively improve the ecological efficiency of industries in the Yangtze River Economic Belt, the following suggestions are made in conjunction with the findings of this paper.

First, strengthen the ecological environment construction of the Yangtze River Economic Belt. Given the current situation that the overall industrial eco-efficiency level of the Yangtze River Economic Belt is low and the development process of industrial ecology is slow, the efforts of ecological badlands management in each region should be increased. Specifically, each province should pay attention to the treatment and restoration of industrial ecosystems, improve the monitoring network and supervision system of the ecological environment quality of the Yangtze River Economic Belt, tackle the weaknesses of water pollution and ecological fragility in the Yangtze River Economic Belt, eliminate backward production capacity, strengthen ecological restoration and treatment, and strive to achieve a win-win situation of green and development in the Yangtze River Economic Belt, and push the high-quality green development of industries in the Yangtze River Economic Belt with green ecology.

Second, reduce the spatial differences in industrial eco-efficiency of 11 provinces and cities in the Yangtze River Economic Belt. Each province should start from its geographical factors, economic and social development status, and industrial eco-efficiency level and analyze specific problems to improve its industrial eco-efficiency. At the same time, strong provinces should help weak provinces and strengthen cooperation between neighboring provinces to promote the joint improvement of industrial eco-efficiency levels in the region. For such high—high agglomeration areas as Shanghai, Jiangsu, and Zhejiang in the downstream region, they should take the initiative to take more ecological responsibilities, promote compelling experiences about environmental management, ecological protection, and green technology to the neighboring provinces with low industrial eco-efficiency level, and give full play to the spillover effect and radiation-driven effect of their industrial eco-development. For the low-low agglomeration areas in the upper and middle reaches of the Yangtze River, they should exert their subjective initiative and seek advice from neighboring provinces with high levels of industrial eco-efficiency around them, for example, by conducting regular study and exchange meetings about ecological governance, learning the management experience, development concepts and control measures of high-level provinces, as well as introducing green eco-technologies from outside the province and abroad, to improve their industrial eco-efficiency levels.

Third, it will deeply promote the synergistic regional development of the Yangtze River Economic Belt. Strengthen cooperation among the 11 provinces, optimize resource allocation, especially water and mineral resources, and form a core region with a driving and radiating effect. Specifically, we can change the traditional resource- and environment-intensive production methods, control energy consumption and total pollution emissions, pay attention to ecological and environmental protection of water resources and mineral resources, and improve the prevention and control mechanisms for the defense of clean skies, water, and soil.

## Supporting information

**S1 Data. For data used in the manuscript, please refer to supporting data.**
(XLSX)

## Author Contributions

**Conceptualization:** Liang Liu.

**Data curation:** Yirui Yang, Xiujuan Gong.

**Investigation:** Xiujuan Gong.

**Project administration:** Pan Jiang.

**Resources:** Yirui Yang.

**Software:** Yuting Zhao, Shu Liu.

**Supervision:** Liang Liu.

**Visualization:** Shu Liu.

**Writing – original draft:** Yuting Zhao, Pan Jiang.

**Writing – review & editing:** Pan Jiang.

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
