## [Decision Letter · Decision Letter 0]

2 Jan 2023

PONE-D-22-31799Study on the measurement of industrial eco-efficiency, spatial distribution and influencing factors in Yangtze River Economic BeltPLOS ONE

Dear Dr. Jiang,

Thank you for submitting your manuscript to PLOS ONE. After careful consideration, we feel that it has merit but does not fully meet PLOS ONE’s publication criteria as it currently stands. Therefore, we invite you to submit a revised version of the manuscript that addresses the points raised during the review process.

We look forward to receiving your revised manuscript.

Kind regards,

Bing Xue, Ph.D.

Academic Editor

PLOS ONE

Journal Requirements:

2. Our internal editors have looked over your manuscript and determined that it is within the scope of our Sustainability and the Circular Economy Call for Papers. The Collection will encompass a diverse and interdisciplinary set of submissions related to sustainability and the circular economy, focusing on production models, business plans, and the contribution of global initiatives to increased sustainability in economic, environmental, and social terms. Additional information can be found on our announcement page: Sustainability and the Circular Economy - PLOS Collections . If you would like your manuscript to be considered for this collection, please let us know in your cover letter and we will ensure that your paper is treated as if you were responding to this call. If you would prefer to remove your manuscript from collection consideration, please specify this in the cover letter.

"LL have been funded by the Strategic Research and Consulting Project of the Chinese Academy of Engineering (Project Numbers: 2021-XY-16), the National Natural Science Foundation of China (Project Numbers: 72004188) and the Sichuan Provincial Science and Technology Program Projects (Project Numbers: 2022JDR0177)."

4. Thank you for stating the following in the Funding Section of your manuscript: 

"LL have been funded by the Strategic Research and Consulting Project of the Chinese Academy of Engineering (Project Numbers: 2021-XY-16), the National Natural Science Foundation of China (Project Numbers: 72004188) and the Sichuan Provincial Science and Technology Program Projects (Project Numbers: 2022JDR0177)."

"LL have been funded by the Strategic Research and Consulting Project of the Chinese Academy of Engineering (Project Numbers: 2021-XY-16), the National Natural Science Foundation of China (Project Numbers: 72004188) and the Sichuan Provincial Science and Technology Program Projects (Project Numbers: 2022JDR0177)."

Reviewers' comments:

Reviewer's Responses to Questions

**Comments to the Author**

1. Is the manuscript technically sound, and do the data support the conclusions?

Reviewer #1: Yes

Reviewer #2: Yes

Reviewer #3: Yes

Reviewer #4: Yes

2. Has the statistical analysis been performed appropriately and rigorously? 

Reviewer #1: Yes

Reviewer #2: Yes

Reviewer #3: Yes

Reviewer #4: Yes

3. Have the authors made all data underlying the findings in their manuscript fully available?

Reviewer #1: No

Reviewer #2: Yes

Reviewer #3: Yes

Reviewer #4: Yes

4. Is the manuscript presented in an intelligible fashion and written in standard English?

Reviewer #1: Yes

Reviewer #2: Yes

Reviewer #3: Yes

Reviewer #4: Yes

5. Review Comments to the Author

Reviewer #1: It is a great honor to review this paper. It is of great significance to study the ecological efficiency of the Yangtze River Economic Belt in China. The structure of this paper is relatively complete; the method is correct; and the conclusion is reasonable. This is a more standardized and standard paper. However, I think it still has a lot of problems to deal with, so the current version is not suitable for publication. I hope that some of my suggestions can help the author revise the paper and improve its quality.

1.The theme, method, data, and conclusion of this paper are not innovative, so I suggest that the author highlight at least one aspect of innovation, otherwise, it will not attract reviewers and readers.

2.The introduction section only mentions China's political and economic background, and although it is important to list some important meetings and conversations, it is far from enough. This is a paper with data from China, but I still suggest that the author put forward a global macro background, which is more attractive to international readers.

3.The introduction section simply enumerates some contents; there is no logical progression, and the current content does not explain this issue clearly.

4.The author needs to make a major revision to the literature review, at present, the logic of this part of the content is quite confused, and I do not understand the author's writing ideas. The citation format of some documents is incorrect, such as lines 123 and 125. The author quoted in the article usually does not use the full name, and the time should be added after the author.

5.Generally, after the introduction or literature review, the innovation and marginal contribution of this paper should be put forward, which is obviously not mentioned by the author.

6.In addition to the existing literature and data, I would like to see the author choose the correct theory to analyze this issue. Unfortunately, the author does not carry out theoretical analysis but directly uses empirical analysis, which I think is incomplete. Therefore, I suggest that the author add "theoretical analysis", which will increase the attractiveness of the article.

7.I suggest that after reporting the results of the empirical model, the author should briefly analyze the reasons for the results.

8.Reviewers and readers pay great attention to the section on "discussion," but it is unreasonable that the author does not compare and discuss the research results with the existing literature. Therefore, I strongly suggest that the author add the "discussion" section before the conclusion.

9. There are many grammatical problems in the article that need to be revised.

If some of my suggestions are incorrect, I hope the author can forgive me, Thank you!

Reviewer #2: The development of the Yangtze River Economic Belt is the main force leading Chinese high-quality development, and it covers 11 provinces and cities with widely varying geographical conditions and uneven levels of development. This paper concentrates on exploring industrial eco-efficiency, thinking at the economic, social and environmental levels, constructing an efficiency index system, analyzing its spatial and temporal evolution characteristics and influencing factors. The manuscript has scientific conception, reasonable results and standard English. I would suggest accepting it after a minor revision. Here are some suggestions:

From the content of the manuscript writing:

Abstract: The abstract section is clearly mentioning the objective, methods, and concise results with sequence.However, the background of the research is not clearly explained.

Introduction: In the introduction section, the author used a three-paragraph writing style to clearly articulate the background, objectives, research gaps and innovations and mapped out the research framework, It’s well written. But lacked the elaboration of Figure 1-Research framework diagram.

Data and methodology:In this section, the author provides a detailed description of the super-efficient SBM method, spatial weight matrix, and Tobit regression used in the manuscript, but please add the relevance and importance of these indicators in the construction of the indicator system.e.g.225.

Analysis：In the analysis section, the author analyzed the temporal and spatial perspectives of industrial eco-efficiency and explored the influencing factors by using Tobit regression with progressive addition of variables. This section is logically clear and complete, and the use of Arcgis and other software to draw line graphs, distribution charts and scatter plots better aids the illustration.However, there are some problems with this section of the table:(1) Table headings need to be concise to capture the focus, some unnecessary expressions can be deleted. Please briefly write the title of Table 2; (2)The variables in Table 5 need to be abbreviated, and the analysis content contains the description of positive and negative indicators, so it is suggested to add them in Table 5.

Conclusions and recommendations: In the conclusions and recommendations section, the author summarized the overall development characteristics, spatial distribution pattern and influencing factors and made recommendations correspondingly, with clear logic.

Avoid grammatical and typo errors and revise the manuscripts for these concerns:

(1)In the Abstract section, please modify the parentheses in the third point to the correct format.

(2)There is a problem with the format of Table 3, please adjust the format and standardize the data to the third decimal place.

(3)Please clarify the spelling, whether it is “Yangtze River Economic Belt” or “Yangtze River Economic Zone”.

Reviewer #3: The manuscript addresses an important and interesting problem-industrial eco-efficiency in Yangtze River Economic Belt. Improving the ecological environment is a major challenge, the author measured the industrial eco-efficiency of the Yangtze River Economic Belt from 2011 to 2020, and evaluated the changes in efficiency at the general level, the time level, the regional level and the provincial level, then explored its spatial distribution pattern, analyzed the influencing factors of industrial eco-efficiency. Overall, the manuscript is well organized and its presentation is good. I would suggest accepting it. However; some minor issues still need to be improved :

(1) Please remember that when a proper noun appears for the first time in an article, the full name of the proper noun must be used.

(2) In the keyword section, there is a typo error, Moran’I is written incorrectly.

(3) In the introduction section, the authors clearly state the background, research gaps and innovations, which is quite standard, but the research objectives should be more clearly defined at the end of the Introduction.

(4) In the literature review section, summarizing the literature related to industrial eco-efficiency in terms of origin, concept definition, measurement dimensions and methodology, but some language expressions need to be refined.e.g.line128-line131.

(5) In the data and methodology section, it is nice to elaborate on the sample area and data sources for this article, however, please add the appropriate references to show the relevance and necessity of doing so.

(6) Using the super-efficient SBM model, the indicators of inputs and outputs are a matter of great concern. For the selection of indicators of industrial eco-efficiency in the Yangtze River Economic Belt, the author starts from environmental, economic and social aspects, so please add appropriate citations to show where you have selected the indicators from.

(7) In subsection 4.1.3 Regional characteristics, the author chose to use Arcgis software to map the industrial eco-efficiency values for 2011, 2015 and 2020 to illustrate their spatial evolution trends. However, there is repetition of words in the analytical expression.

(8) In subsection 4.2.2 Local autocorrelation, a scatter plot is drawn to divide the 11 provinces and cities of the Yangtze River Economic Belt into four regions for analysis, which is very clear, but please note the expression of proper nouns.e.g.line422 line431 line440 line450.

(9) In subsection 4.2.3, the author consider the shortcomings of the super-efficient SBM and choose to apply tobit regression to analyze the factors influencing the industrial eco-efficiency, constructing six regression models, please explain the meaning of each variable in the formula for readers to have a clearer understanding of the model.

Reviewer #4: This manuscript uses the super-efficient SBM model to construct an index system from economic, environmental, and resource factors to scientifically measure the industrial eco-efficiency of the Yangtze River Economic Belt, and reveals the spatial differences in industrial eco-efficiency of different provincial units in the Yangtze River Economic Belt, and explores the influencing factors of industrial eco-efficiency in the Yangtze River Economic Belt. It enriches the research on industrial ecology and fills the gap of research targeting provincial research scale. Therefore, I would suggest accepting it after the following minor concerns are addressed：

1. In the abstract, the authors explain the background of the study and the main contents and conclusions of the study, but a sentence needs to be added to the abstract to clearly explain the necessity of this study.

2. In the introduction the authors describe the background and significance of the study, but a sentence or two needs to be added to clearly explain the innovation and specific contribution of the study.

3.It is reasonable to select input indicators from three aspects: economy, environment and resources, but some references need to be added in this paragraph (line231-line239).

4.The text annotations below the table are used to supplement the content of the table, but the annotations about the table please indicate for example lines 267 and 490

5.The data in the table are calculated from the model and the contents of the table should be explained to facilitate understanding. For example, the meaning of the values represented by each column in Table 4.

6.The punctuation in the article is consistent with the need to make reasonable use of it so as not to affect the reading. Please check the punctuation throughout the article for correctness, such as the semicolon in line 339.

7.The article uses a table to show the names and units of the indicators, which is very clear and concise. However, it is important to note that the case of the words in each phrase in the table is consistent. Please check the spelling of the words in each table to make sure that the case in each table is consistent. e.g. 'billion cubic meters' in Tab.1.

6. PLOS authors have the option to publish the peer review history of their article (what does this mean?). If published, this will include your full peer review and any attached files.

Reviewer #1: No

Reviewer #2: No

Reviewer #3: No

Reviewer #4: No

---

## [Author Response · Author response to Decision Letter 0]

21 Feb 2023

Thank you for your valuable comments. For a detailed response, please check the attachment!

Sincerely, thank you for your help.

---

## [Decision Letter · Decision Letter 1]

21 Mar 2023

Study on the measurement of industrial eco-efficiency, spatial distribution and influencing factors in Yangtze River Economic Belt

PONE-D-22-31799R1

Dear Dr. Jiang,

We’re pleased to inform you that your manuscript has been judged scientifically suitable for publication and will be formally accepted for publication once it meets all outstanding technical requirements.

Kind regards,

Bing Xue, Ph.D.

Academic Editor

PLOS ONE

Additional Editor Comments (optional):

Reviewers' comments:

Reviewer's Responses to Questions

**Comments to the Author**

1. If the authors have adequately addressed your comments raised in a previous round of review and you feel that this manuscript is now acceptable for publication, you may indicate that here to bypass the “Comments to the Author” section, enter your conflict of interest statement in the “Confidential to Editor” section, and submit your "Accept" recommendation.

Reviewer #1: All comments have been addressed

2. Is the manuscript technically sound, and do the data support the conclusions?

Reviewer #1: Yes

3. Has the statistical analysis been performed appropriately and rigorously? 

Reviewer #1: Yes

4. Have the authors made all data underlying the findings in their manuscript fully available?

Reviewer #1: Yes

5. Is the manuscript presented in an intelligible fashion and written in standard English?

Reviewer #1: Yes

6. Review Comments to the Author

Reviewer #1: The author has addressed all the comments I have made and I recommend that the current version of the manuscript be accepted and published. Of course, the final decision rests with the editor.

7. PLOS authors have the option to publish the peer review history of their article (what does this mean?). If published, this will include your full peer review and any attached files.

Reviewer #1: No

---

## [Editor Report · Acceptance letter]

27 Mar 2023

PONE-D-22-31799R1 

Study on the measurement of industrial eco-efficiency, spatial distribution and influencing factors in Yangtze River Economic Belt 

Dear Dr. Jiang:

I'm pleased to inform you that your manuscript has been deemed suitable for publication in PLOS ONE. Congratulations! Your manuscript is now with our production department. 

Kind regards, 

on behalf of

Professor Bing Xue 

Academic Editor

PLOS ONE